# Diabetes Mellitus and Cardiovascular Risk Assessment in Mothers with a History of Gestational Diabetes Mellitus Based on Postpartal Expression Profile of MicroRNAs Associated with Diabetes Mellitus and Cardiovascular and Cerebrovascular Diseases

**DOI:** 10.3390/ijms21072437

**Published:** 2020-03-31

**Authors:** Ilona Hromadnikova, Katerina Kotlabova, Lenka Dvorakova, Ladislav Krofta

**Affiliations:** 1Department of Molecular Biology and Cell Pathology, Third Faculty of Medicine, Charles University, 10000 Prague, Czech Republic; katerina.kotlabova@lf3.cuni.cz (K.K.); lenka.dvorakova@lf3.cuni.cz (L.D.); 2Institute for the Care of the Mother and Child, Third Faculty of Medicine, Charles University, 14700 Prague, Czech Republic; ladislav.krofta@upmd.eu

**Keywords:** cardiovascular/cerebrovascular diseases, cardiovascular risk, gestational diabetes mellitus, microRNA expression, mothers, pregnancy complications, primary prevention, screening

## Abstract

Mothers with a history of gestational diabetes mellitus (GDM) have an increased risk of developing diabetes in the future and a lifelong cardiovascular risk. Postpartal expression profile of cardiovascular/cerebrovascular disease associated microRNAs was assessed 3–11 years after the delivery in whole peripheral blood of young and middle-aged mothers with a prior exposure to GDM with the aim to identify a high-risk group of mothers at risk of later development of diabetes mellitus and cardiovascular/cerebrovascular diseases who would benefit from implementation of early primary prevention strategies and long-term follow-up. The hypothesis of the assessment of cardiovascular risk in women was based on the knowledge that a series of microRNAs play a role in the pathogenesis of diabetes mellitus and cardiovascular/cerebrovascular diseases. Abnormal expression profile of multiple microRNAs was found in women with a prior exposure to GDM (miR-1-3p, miR-16-5p, miR-17-5p, miR-20a-5p, miR-20b-5p, miR-21-5p, miR-23a-3p, miR-24-3p, miR-26a-5p, miR-29a-3p, miR-100-5p, miR-103a-3p, miR-125b-5p, miR-126-3p, miR-130b-3p, miR-133a-3p, miR-143-3p, miR-145-5p, miR-146a-5p, miR-181a-5p, miR-195-5p, miR-199a-5p, miR-221-3p, miR-342-3p, miR-499a-5p, and-miR-574-3p). Postpartal combined screening of miR-1-3p, miR-16-5p, miR-17-5p, miR-20b-5p, miR-21-5p, miR-23a-3p, miR-26a-5p, miR-29a-3p, miR-103a-3p, miR-133a-3p, miR-146a-5p, miR-181a-5p, miR-195-5p, miR-199a-5p, miR-221-3p, and miR-499a-5p showed the highest accuracy for the identification of mothers with a prior exposure to GDM at a higher risk of later development of cardiovascular/cerebrovascular diseases (AUC 0.900, *p* < 0.001, sensitivity 77.48%, specificity 93.26%, cut off >0.611270413). It was able to identify 77.48% mothers with an increased cardiovascular risk at 10.0% FPR. Any of changes in epigenome (upregulation of miR-16-5p, miR-17-5p, miR-29a-3p, and miR-195-5p) that were induced by GDM-complicated pregnancy are long-acting and may predispose mothers affected with GDM to later development of diabetes mellitus and cardiovascular/cerebrovascular diseases. In addition, novel epigenetic changes (upregulation of serious of microRNAs) appeared in a proportion of women that were exposed to GDM throughout the postpartal life. Likewise, a previous occurrence of either GH, PE, and/or FGR, as well as a previous occurrence of GDM, is associated with the upregulation of miR-1-3p, miR-17-5p, miR-20a-5p, miR-20b-5p, miR-29a-3p, miR-100-5p, miR-125b-5p, miR-126-3p, miR-130b-3p, miR-133a-3p, miR-143-3p, miR-145-5p, miR-146a-5p, miR-181a-5p, miR-199a-5p, miR-221-3p, and miR-499a-5p. On the other hand, upregulation of miR-16-5p, miR-21-5p, miR-23a-3p, miR-24-3p, miR-26a-5p, miR-103a-3p, miR-195-5p, miR-342-3p, and miR-574-3p represents a unique feature of aberrant expression profile of women with a prior exposure to GDM. Screening of particular microRNAs may stratify a high-risk group of mothers with a history of GDM who might benefit from implementation of early primary prevention strategies.

## 1. Introduction

Recently, we demonstrated that history of pregnancy-related complications, such as gestational hypertension (GH), preeclampsia (PE), and/or fetal growth restriction (FGR), was associated with postpartum epigenetic changes characteristic for cardiovascular and cerebrovascular diseases [1,2,3].

We showed that previous occurrence of either GH, PE, and/or FGR might predispose a proportion of women to later development of cardiovascular and cerebrovascular diseases due to the presence of alterations in the expression of cardiovascular and cerebrovascular disease-associated microRNAs in their whole peripheral blood [1,2,3]. 

Gestational diabetes mellitus (GDM) is defined as carbohydrate intolerance that develops during pregnancy, usually during second or third trimester of gestation [4]. Women with GDM have an increased risk of developing diabetes (predominantly type 2 diabetes) later in life. It is estimated that up to 70% of women with GDM will develop diabetes within 22–28 years after pregnancy [4,5,6,7]. 

The incidence of GDM increases with the same risk factors seen for type 2 diabetes such as obesity, sedentary lifestyle, and increasing reproductive age of women [4,5,6,7]. 

The initial criteria for diagnosis of GDM were established more than 40 years ago to identify women at a high risk for development of diabetes after pregnancy [8,9,10]. The two-step screening approach to testing for GDM is currently recommended by The American College of Obstetricians and Gynecologists [4] and The International Association of Diabetes and Pregnancy Study Groups (IADPSG) [10]. Overall, using the proposed IADPSG criteria, 18% of pregnant women in the United States are identified as having GDM [10].

The goal of the current study was to evaluate the risk of later development of diabetes mellitus, and cardiovascular and cerebrovascular diseases based on an epigenetic profile of cardiovascular/cerebrovascular disease associated microRNAs in whole peripheral blood of young and middle-aged mothers with a history of gestational diabetes mellitus (GDM) 3–11 years after the delivery. The hypothesis of the assessment of cardiovascular risk in women was based on the knowledge that a series of microRNAs play a role in the pathogenesis of diabetes mellitus and cardiovascular/cerebrovascular diseases (Table 1) [11,12,13,14,15,16,17,18,19,20,21,22,23,24,25,26,27,28,29,30,31,32,33,34,35,36,37,38,39,40,41,42,43,44,45,46,47,48,49,50,51,52,53,54,55,56,57,58,59,60,61,62,63,64,65,66,67,68,69,70,71,72,73,74,75,76,77,78,79,80,81,82,83,84,85,86,87,88,89,90,91,92,93,94,95,96,97,98,99,100,101,102,103,104,105,106,107,108,109,110,111,112,113,114,115,116,117,118,119,120,121,122,123,124,125,126,127,128,129,130,131,132,133,134,135,136,137,138,139,140,141,142,143,144,145,146,147,148,149,150,151,152,153,154,155,156,157,158,159,160,161,162,163,164,165,166,167,168,169,170].

Postpartal epigenetic profiling of microRNAs (miR-1-3p, miR-16-5p, miR-17-5p, miR-20a-5p, miR-20b-5p, miR-21-5p, miR-23a-3p, miR-24-3p, miR-26a-5p, miR-29a-3p, miR-92a-3p, miR-100-5p, miR-103a-3p, miR-125b-5p, miR-126-3p, miR-130b-3p, miR-133a-3p, miR-143-3p, miR-145-5p, miR-146a-5p, miR-155-5p, miR-181a-5p, miR-195-5p, miR-199a-5p, miR-210-3p, miR-221-3p, miR-342-3p, miR-499a-5p, and miR-574-3p) known to be involved in the onset of insulin resistance and diabetes, dyslipidemia, hypertension, vascular inflammation, atherosclerosis, angiogenesis, coronary artery disease, myocardial infarction and heart failure, stroke, intracranial aneurysm, pulmonary arterial hypertension, and peripartum cardiomyopathy was the subject of our interest.

MicroRNAs, small noncoding RNAs, are known to regulate gene expression at the posttranscriptional level by blocking translation or degrading of target messenger RNA [171]. 

To the best of our present knowledge, no study on expression profiling of microRNAs associated with diabetes mellitus, and cardiovascular and cerebrovascular diseases in whole peripheral blood of mothers after pregnancies affected by gestational diabetes mellitus has been carried out. 

## 2. Results

MicroRNA gene expression was compared between mothers after normal and GDM-complicated pregnancies in maternal whole peripheral blood 3 to 11 years postpartum. Just the results, when the complicated cases reached a statistical significance, are presented below.

### 2.1. Expression Profile of MicroRNAs Associated with Diabetes Mellitus and Cardiovascular/Cerebrovascular Diseases in Mothers after GDM Pregnancies 

The expression of miR-1-3p (*p* < 0.001), miR-16-5p (*p* < 0.001), miR-17-5p (*p* < 0.001), miR-20a-5p (*p* < 0.001), miR-20b-5p (*p* < 0.001), miR-21-5p (*p* < 0.001), miR-23a-3p (*p* < 0.001), miR-24-3p (*p* = 0.007), miR-26a-5p (*p* < 0.001), miR-29a-3p (*p* < 0.001), miR-100-5p (*p* < 0.001), miR-103a-3p (*p* < 0.001), miR-125b-5p (*p* < 0.001), miR-126-3p (*p* < 0.001), miR-130b-3p (*p* < 0.001), miR-133a-3p (*p* < 0.001), miR-143-3p (*p* < 0.001), miR-145-5p (*p* = 0.002), miR-146a-5p (*p* < 0.001), miR-181a-5p (*p* < 0.001), miR-195-5p (*p* < 0.001), miR-199a-5p (*p* < 0.001), miR-221-3p (*p* < 0.001), miR-342-3p (*p* = 0.003), miR-499a-5p (*p* < 0.001), and miR-574-3p (*p* < 0.001) differed significantly between the control group and women previously affected with GDM. These microRNAs were upregulated in GDM-affected patients (Figure 1 and Figure 2). 

The ROC curve analysis confirmed a significant upregulation of these particular microRNAs for mothers with a history of GDM when the comparison to the controls was performed (Figure 1). 

The very good sensitivity at 10.0% FPR for miR-1-3p (43.24%), miR-16-5p (20.72%), miR-17-5p (21.62%), miR-20b-5p (38.74%), miR-21-5p (28.83%), miR-23a-3p (17.12%), miR-26a-5p (20.72%), miR-29a-3p (21.62%), miR-103a-3p (30.63%), miR-133a-3p (18.92%), miR-146a-5p (17.12%), miR-181a-5p (31.53%), miR-195-5p (16.22%), miR-199a-5p (15.32%), miR-221-3p (31.53%), and miR-499a-5p (28.83%) is found (Figure 2). 

However, despite the upregulation of microRNAs, the poor sensitivity at 10.0% FPR for miR-20a-5p (10.81%), miR-24-3p (0.90%), miR-100-5p (13.51%), miR-125b-5p (9.91%), miR-126-3p (9.91%), miR-130b-3p (3.60%), miR-143-3p (13.51%), miR-145-5p (7.21%), miR-342-3p (1.8%), and miR-574-3p (10.81%) was detected, therefore, these particular microRNAs were not further used for diabetes mellitus and cardiovascular risk assessment in mothers previously affected with GDM (Figure 2).

Combined screening of miR-1-3p, miR-16-5p, miR-17-5p, miR-20b-5p, miR-21-5p, miR-23a-3p, miR-26a-5p, miR-29a-3p, miR-103a-3p, miR-133a-3p, miR-146a-5p, miR-181a-5p, miR-195-5p, miR-199a-5p, miR-221-3p, and miR-499a-5p showed the highest accuracy for mothers with a prior exposure to GDM (AUC 0.900, *p* < 0.001, sensitivity 77.48%, specificity 93.26%, cut off > 0.611270413). It was able to identify 77.48% of mothers with an increased cardiovascular risk at 10.0% FPR (Figure 3). 

### 2.2. Expression Profile of MicroRNAs Associated with Diabetes Mellitus and Cardiovascular/Cerebrovascular Diseases in Mothers after GDM Pregnancies with Regard to the Treatment Strategies (Diet Only and/or Diet and Therapy) 

Concurrently, it was observed that the expression of miR-1-3p (*p* < 0.001, *p* = 0.013), miR-16-5p (*p* < 0.001, *p* = 0.057), miR-17-5p (*p* < 0.001, *p* = 0.035), miR-20b-5p (*p* < 0.001, *p* = 0.002), miR-21-5p (*p* < 0.001, *p* < 0.001), miR-23a-3p (*p* < 0.001, *p* = 0.096), miR-26a-5p (*p* < 0.001, *p* = 0.002), miR-29a-3p (*p* < 0.001, *p* < 0.001), miR-103a-3p (*p* < 0.001, *p* = 0.001), miR-146a-5p (*p* < 0.001, *p* = 0.019), miR-181a-5p (*p* < 0.001, *p* = 0.003), miR-195-5p (*p* < 0.001, *p* < 0.001), miR-199a-5p (*p* < 0.001, *p* = 0.002), miR-221-3p (*p* < 0.001, *p* < 0.001), and miR-499a-5p (*p* < 0.001, *p* < 0.001) differed significantly or showed a trend toward statistical significance between the groups of mothers affected with GDM regardless of the treatment strategies (diet only and/or the combination of diet and therapy) and the controls (Figure 4). Nevertheless, miR-133a-3p was upregulated just in the group of patients previously affected with GDM on diet only (*p* = 0.001) (Figure 4). 

Due to a low number of women on the diet and therapy (*n* = 18) in our studied group of patients previously affected with GDM, we decided not to perform ROC curve analysis for particular groups of patients regarding the GDM treatment strategies (diet only and/or combination of diet and therapy).

No difference in microRNA expression profiles was observed between the groups of mothers with a history of GDM on diet only and on the combination of diet and therapy (*p* = 1.0 for nearly all examined microRNAs).

### 2.3. Information on MicroRNA-Gene-Biological Pathway Interactions

The extensive file of predicted targets of all microRNAs aberrantly expressed in whole peripheral blood of mothers with a history of GDM indicates that a large group of genes is involved in biological pathways related to insulin signaling, type 1 diabetes mellitus, and type 2 diabetes mellitus (Table 2, Table 3 and Table 4).

## 3. Discussion

Postpartal expression profile of cardiovascular/cerebrovascular disease associated microRNAs was assessed 3–11 years after the delivery in whole peripheral blood of young and middle-aged mothers with a prior exposure to GDM with the aim to identify a high-risk group of mothers at risk of later development of diabetes mellitus and cardiovascular and cerebrovascular diseases who would benefit from implementation of early primary prevention strategies and long-term follow-up. The hypothesis of the assessment of cardiovascular risk in women was based on the knowledge that a serious of microRNAs play a role in the pathogenesis of diabetes mellitus and cardiovascular/cerebrovascular diseases (Table 1) [11,12,13,14,15,16,17,18,19,20,21,22,23,24,25,26,27,28,29,30,31,32,33,34,35,36,37,38,39,40,41,42,43,44,45,46,47,48,49,50,51,52,53,54,55,56,57,58,59,60,61,62,63,64,65,66,67,68,69,70,71,72,73,74,75,76,77,78,79,80,81,82,83,84,85,86,87,88,89,90,91,92,93,94,95,96,97,98,99,100,101,102,103,104,105,106,107,108,109,110,111,112,113,114,115,116,117,118,119,120,121,122,123,124,125,126,127,128,129,130,131,132,133,134,135,136,137,138,139,140,141,142,143,144,145,146,147,148,149,150,151,152,153,154,155,156,157,158,159,160,161,162,163,164,165,166,167,168,169,170].

As expected, the expression profile of microRNAs differed between mothers affected with GDM and controls. Abnormal expression profile of multiple microRNAs was found in mothers with a history of GDM (26/29 studied microRNAs: miR-1-3p, miR-16-5p, miR-17-5p, miR-20a-5p, miR-20b-5p, miR-21-5p, miR-23a-3p, miR-24-3p, miR-26a-5p, miR-29a-3p, miR-100-5p, miR-103a-3p, miR-125b-5p, miR-126-3p, miR-130b-3p, miR-133a-3p, miR-143-3p, miR-145-5p, miR-146a-5p, miR-181a-5p, miR-195-5p, miR-199a-5p, miR-221-3p, miR-342-3p, miR-499a-5p, and miR-574-3p).

Nevertheless, when the ROC curve analysis was performed, only 16/26 microRNAs (miR-1-3p, miR-16-5p, miR-17-5p, miR-20b-5p, miR-21-5p, miR-23a-3p, miR-26a-5p, miR-29a-3p, miR-103a-3p, miR-133a-3p, miR-146a-5p, miR-181a-5p, miR-195-5p, miR-199a-5p, miR-221-3p, and miR-499a-5p) with aberrant postpartal expression profile in whole peripheral blood of mothers with a prior exposure to GDM showed a higher sensitivity, ranging from 15.32% to 43.24%, at 10.0% FPR. 

Screening based on the combination of these particular microRNAs was superior over using individual microRNAs, since it showed the highest accuracy for mothers with a history of GDM (AUC 0.900, *p* < 0.001, sensitivity 77.48%, specificity 93.26%, cut off > 0.611270413). It was able to identify 77.48% mothers with an increased cardiovascular risk at 10.0% FPR.

Subsequently, epigenetic profile was compared between groups with regard to the treatment strategy (GDM on diet only, GDM on the combination of diet and therapy). The upregulation or trend towards upregulation of almost all studied microRNAs (with the exception of miR-133a-3p) was present in both groups of mothers affected with GDM regardless of the treatment strategies. 

Previously, we demonstrated that pregnancy-related complications, such as gestational hypertension, preeclampsia, and/or fetal growth restriction, were associated with postpartum alterations in gene expression of cardiovascular/cerebrovascular disease-associated microRNAs [1,2,3]. 

Likewise, a previous occurrence of GH, as well as a previous occurrence of GDM, was associated with the upregulation of miR-20a-5p, miR-143-3p, miR-146a-5p, miR-181a-5p, miR-199a-5p, miR-221-3p, and miR-499a-5p [1,2,3]. 

Moreover, the compilation of data resulting from our previous and current studies showed that the upregulation of miR-17-5p, miR-20b-5p, miR-29a-3p, and miR-126-3p was a mutual phenomenon observed in women with a prior exposure to GH, severe PE, and/or GDM [1,2,3]. 

In addition, a history of GH, early PE, and/or GDM was associated with postpartal upregulation of miR-1-3p and miR-17-5p. Parallel, women affected with GH and/or late PE showed similar postpartal expression profile (upregulation of miR-17-5p, miR-20b-5p, and miR-29a-3p) as women with a history of GDM [1,2,3]. 

Alike, as in women affected with severe PE and/or early PE, upregulation of miR-133a-3p was also present in women affected with GDM. Moreover, a history of severe PE and/or GDM was associated with upregulation of miR-130b-3p [1,2,3]. 

Furthermore, women with a history of GDM demonstrated upregulation of miR-100-5p, miR-125b-5p, miR-133a-3p, and miR-145-5p as women with prior exposure to PE and/or FGR with abnormal Doppler parameters [1,2,3]. 

On the other hand, our study revealed that upregulation of miR-16-5p, miR-21-5p, miR-23a-3p, miR-24-3p, miR-26a-5p, miR-103a-3p, miR-195-5p, miR-342-3p, and miR-574-3p represented a unique feature of aberrant expression profile of women with a prior exposure to GDM.

Interestingly, interaction network analysis via Cytoscale V.3.6.1 revealed that miR-145-5p together with miR-875-5p are upregulated microRNAs that target the most genes in gestational diabetes mellitus [125]. 

In addition, increased expression of miR-126-3p and miR-130b-3p was observed in human umbilical vein endothelial cells (HUVECs) derived from GDM patients [98]. 

Moreover, a set of circulating microRNAs associated with cardiovascular/cerebrovascular diseases (miR-16-5p, miR-17-5p, miR-20a-5p, miR-29a-3p, miR-125b-5p, and miR-195-5p) was reported by several independent studies [20,21,80,92,143,172,173] to be upregulated during various gestational ages in serum or plasma of mothers with already-diagnosed GDM or mothers who were destined to develop GDM later during pregnancy. The data resulting from these prenatally ongoing studies and interaction network analysis [20,21,80,92,98,125,143,172,173] may support our current finding referring to upregulated expression profile of these particular microRNAs in whole peripheral blood of women with a prior exposure to GDM. 

In addition, from the portfolio of microRNAs we tested, just miR-16-5p, miR-17-5p, miR-20a-5p, and miR-29a were identified by other investigators as the most promising biomarkers of GDM or the best predictors of GDM, respectively [21,173], which is consistent with our current finding, since during the course of postpartum screening, miR-16-5p (20.72%), miR-17-5p (21.62%), and miR-29a-3p (21.62%) upregulation persisted in a larger proportion of women with a prior exposure to GDM. 

In compliance with our study, which showed a low sensitivity of miR-20a-5p (10.81%) to select women with a higher cardiovascular risk after GDM gestation based on its upregulated profile, miR-20a-5p also added little value as the predictor of GDM due to its low predictive value based on its upregulated profile (ROC, AUC just 0.740) [21,173].

This study demonstrated that the dysregulation of at least four microRNAs (miR-16-5p, miR-17-5p, miR-29a-3p, and miR-195-5p) induced by GDM-complicated pregnancy in maternal circulation (plasma or serum) is present as well in circulation (whole peripheral blood) of a larger proportion of mothers (20.72%, 21.62%, 21.62%, 16.22%) with hindsight (3 to 11 years after the delivery) after the exposure to GDM. 

It is obvious that changes in epigenome induced by GDM, which are persistently present in maternal circulation, may cause later development of diabetes mellitus and cardiovascular and cerebrovascular diseases in women with a prior exposure to GDM. 

However, most of microRNAs, which were the subjects of our interest, have not yet been observed by other investigators to be dysregulated in maternal circulation (peripheral blood, plasma, or serum samples) of women with clinically established GDM or before the onset of GDM [20,21,37,80,92,143,149,172,173,174,175,176]. So, it is feasible that epigenetic profiles of a series of microRNAs have also been changing with time by force of various circumstances as a result of the interaction between genetic and environmental factors [177].

## 4. Materials and Methods 

### 4.1. Participants 

The study included a prospectively collected cohort of Caucasian mothers with a history of GDM (*n* = 111) and age-matched mothers after normal course of gestation (*n* = 89). In-person visit was conducted 3–11 years after the pregnancy ended. Of the 111 GDM pregnancies, 93 were on diet only and 18 were on the combination of diet and therapy (17 patients required insulin administration and in 1 patient metformin, an oral hypoglycemic agent, was prescribed).

The clinical characteristics of mothers after GDM-complicated pregnancies are presented in Table 5, Table 6 and Table 7.

Gestational diabetes mellitus is defined as any degree of glucose intolerance with onset or first recognition during pregnancy [10,178,179]. The International Association of Diabetes and Pregnancy Study Groups’ (IADPSG) recommendations on the diagnosis and classification of hyperglycemia in pregnancy were followed [10]. The first screening phase detects, during the first trimester of gestation, women with overt diabetes (fasting plasma glucose level is ≥7.0 mmol/L) and women with GDM (fasting plasma glucose level ≥ 5.1 mmol/L < 7.0 mmol/L). The second screening phase, 2-h 75-g OGTT, at 24–28 weeks of gestation is done in all women not previously found to have overt diabetes or GDM, and identifies GDM if fasting plasma glucose level is ≥5.1 mmol/L or 1-h plasma glucose is ≥10.0 mmol/L or 2-h plasma glucose is ≥8.5 mmol/L [10].

Patients demonstrating other pregnancy-related complications, such as gestational hypertension, preeclampsia, fetal growth restriction, premature rupture of membranes, in utero infections, fetal anomalies or chromosomal abnormalities, and fetal demise in utero or stillbirth, were not involved in the study. 

Written informed consent was provided for all participants included in the study. The study was approved by the Ethics Committee of the Institute for the Care of the Mother and Child, Prague, Czech Republic (grant no. AZV 16-27761A, long-term monitoring of complex cardiovascular profile in the mother, fetus, and offspring descending from pregnancy-related complications, date of approval: 28 May 2015) and by the Ethics Committee of the Third Faculty of Medicine, Prague, Czech Republic (grant no. AZV 16-27761A, long-term monitoring of complex cardiovascular profile in the mother, fetus, and offspring descending from pregnancy-related complications, date of approval: 27 March 2014).

### 4.2. Processing of Samples

Homogenized cell lysates were prepared immediately after collection of whole peripheral blood samples (EDTA tubes, 200 µL) using QIAamp RNA Blood Mini Kit (Qiagen, Hilden, Germany, no: 52304). 

Total RNA was extracted from homogenized cell lysates stored at −80 °C using a mirVana microRNA Isolation kit (Ambion, Austin, USA, no: AM1560) and followed by an enrichment procedure for small RNAs. To minimize DNA contamination, the eluted RNA was treated for 30 min at 37 °C with 5 µL of DNase I (Thermo Fisher Scientific, CA, USA, no: EN0521). A RNA fraction highly enriched in short RNAs (< 200 nt) was obtained. The concentration and quality of RNA was assessed using a NanoDrop ND-1000 spectrophotometer (NanoDrop Technologies, USA). If the A(260/280) absorbance ratio of isolated RNA was 1.8–2.0 and the A(260/230) absorbance ratio was greater than 1.6, the RNA fraction was pure and used for the consecutive analysis. 

### 4.3. Reverse Transcriptase Reaction 

Individual microRNAs were reverse transcribed into complementary DNA (cDNA) in a total reaction volume of 10 µL using microRNA-specific stem-loop RT primers, components of TaqMan MicroRNA Assays (Table 8), and TaqMan MicroRNA Reverse Transcription Kit (Applied Biosystems, Branchburg, USA, no: 4366597). Reverse transcriptase reactions were performed with the following thermal cycling parameters: 30 min at 16 °C, 30 min at 42 °C, 5 min at 85 °C, and then held at 4 °C using a 7500 Real-Time PCR system (Applied Biosystems, Branchburg, USA).

### 4.4. Relative Quantification of MicroRNAs by Real-Time PCR 

The cDNA (3 µL) was mixed with specific TaqMan MGB probes and primers (TaqMan MicroRNA Assay, Applied Biosystems, Branchburg, USA), and the components of the TaqMan Universal PCR Master Mix (Applied Biosystems, Branchburg, USA, no: 4318157). A total reaction volume was 15 µL. TaqMan PCR conditions were set up as described in the TaqMan guidelines for a 7500 Real-Time PCR system. All PCRs were performed in duplicates with the involvement of multiple negative controls such as NTC (water instead of cDNA sample), NAC (nontranscribed RNA samples), and genomic DNA (isolated from equal biological samples), which did not generate any signal during PCR reactions. The samples were considered positive if the amplification signal occurred at Ct < 40 (before the 40th threshold cycle). 

The expression of particular microRNA was determined using the comparative Ct method [180] relative to normalization factor (geometric mean of two selected endogenous controls) [181]. Two noncoding small nucleolar RNAs (RNU58A and RNU38B) were optimal for qPCR data normalization in this setting. They demonstrated equal expression between children descending from normal and complicated pregnancies. RNU58A and RNU38B also served as positive controls for successful extraction of RNA from all samples and were used as internal controls for variations during the preparation of RNA, cDNA synthesis, and real-time PCR. 

A reference sample, RNA fraction highly enriched for small RNAs isolated from the fetal part of one randomly selected placenta derived from gestation with normal course, was used throughout the study for relative quantification. 

### 4.5. Statistical Analysis

Data normality was assessed using the Shapiro–Wilk test [182]. Since our experimental data did not follow a normal distribution, microRNA levels were compared between groups using the Kruskal–Wallis one-way analysis of variance with post hoc test for the comparison among multiple groups. The significance level was established at a *p*-value of p < 0.05. 

Receivers operating characteristic (ROC) curves were constructed to calculate the area under the curve (AUC) and the best cut-off point for particular microRNA was used in order to calculate the respective sensitivity at 90.0% specificity (MedCalc Software bvba, Ostend, Belgium). For every possible threshold or cut-off value, the MedCalc program reports the sensitivity, specificity, likelihood ratio positive (LR+), likelihood ratio negative (LR−).

To select the optimal combinations of microRNA biomarkers, logistic regression was used (MedCalc Software bvba, Ostend, Belgium). The logistic regression procedure allowed us to analyses the relationship between one dichotomous dependent variable and one or more independent variables. Another method to evaluate the logistic regression model makes use of ROC curve analysis. In this analysis, the power of the model’s predicted values to discriminate between positive and negative cases is quantified by the area under the ROC curve (AUC). To perform a full ROC curve analysis, the predicted probabilities are first saved and next used as a new variable in ROC curve analysis. The dependent variable used in logistic regression then acts as the classification variable in the ROC curve analysis dialog box.

Box plots encompassing the median (dark horizontal line) of log-normalized gene expression values for particular microRNAs were generated using Statistica software (version 9.0; StatSoft, Inc., USA). The upper and lower limits of the boxes represent the 75th and 25th percentiles, respectively. The upper and lower whiskers indicate the maximum and minimum values that are no more than 1.5 times the span of the interquartile range (range of the values between the 25th and the 75th percentiles). Outliers are marked by circles and extremes by asterisks. 

### 4.6. Information on MicroRNA-Gene-Biological Pathway Interactions

MiRWalk database (available: http://www.umm.uni-heidelberg.de/apps/zmf/mirwalk/) and the Predicted Target module were used to provide information on predicted targets of those microRNAs that have been found to be dysregulated in whole peripheral blood of patients with a history of GDM. Only those predicted targets involved in particular human biological pathways (insulin signaling pathway, type 1 diabetes mellitus pathway, and type 2 diabetes mellitus pathway) are reported [183]. 

MiRWalk is a comprehensive database that provides information on microRNAs from human, mouse, and rat on their predicted and/or validated target genes. MiRWalk2.0 not only documents miRNA binding sites within the complete sequence of a gene, but also combines this information with a comparison of binding sites resulting from 12 existing miRNA-target prediction programs (DIANA-microTv4.0, DIANA-microT-CDS, miRanda, mirBridge, miRDB4.0, miRmap, miRNAMap, DoRiNA, PicTar2, PITA, RNA22v2, RNAhybrid2.1, and Targetscan6.2) to build novel comparative platforms of binding sites for the promoter (4 prediction datasets), cds (5 prediction datasets), 5’- (5 prediction datasets), and 3’-UTR (13 prediction datasets) regions. Information on miRNA-target interactions on 2035 disease ontologies (DO), 6727 human phenotype ontologies (HPO) and 4980 OMIM disorders is available. It provides possible interactions between microRNAs and genes associated with 597 KEGG, 456 Panther, and 522 Wiki pathways. 

## 5. Conclusions

In conclusion, epigenetic changes characteristic for diabetes mellitus and cardiovascular/cerebrovascular diseases are also present in whole peripheral blood of a proportion of mothers with a history of GDM. Likewise, a previous occurrence of either GH, PE, and/or FGR, as well as a previous occurrence of GDM, is associated with the upregulation of miR-1-3p, miR-17-5p, miR-20a-5p, miR-20b-5p, miR-29a-3p, miR-100-5p, miR-125b-5p, miR-126-3p, miR-130b-3p, miR-133a-3p, miR-143-3p, miR-145-5p, miR-146a-5p, miR-181a-5p, miR-199a-5p, miR-221-3p, and miR-499a-5p.

On the other hand, upregulation of miR-16-5p, miR-21-5p, miR-23a-3p, miR-24-3p, miR-26a-5p, miR-103a-3p, miR-195-5p, miR-342-3p, and miR-574-3p represents a unique feature of aberrant expression profile of women with a prior exposure to GDM.

Any changes in epigenome (upregulation of miR-16-5p, miR-17-5p, miR-29a-3p, and miR-195-5p) that were induced by GDM-complicated pregnancy are long-acting and may predispose mothers affected with GDM to later development of diabetes mellitus and cardiovascular/cerebrovascular diseases. In addition, novel epigenetic changes (upregulation of a series of microRNAs) appeared in a proportion of women that were exposed to GDM throughout the postpartal life.

Overall, previous occurrence of GDM, therefore, predisposes a proportion of affected individuals to later development of diabetes mellitus and cardiovascular/cerebrovascular diseases. Screening of particular microRNAs may stratify a high-risk group of mothers with a history of GDM that might benefit from implementation of early primary prevention strategies.

Consecutive large-scale studies, including the groups of mothers on diet only and on the combination of diet and therapy, are needed to verify the findings resulting from this particular pilot study.

## 6. Patents

National patent granted—Industrial Property Office, Czech Republic (Patent no. 308178).

International patent filed—Industrial Property Office, Czech Republic (PCT/CZ2019/050051).

## Figures and Tables

**Figure 1 ijms-21-02437-f001:**
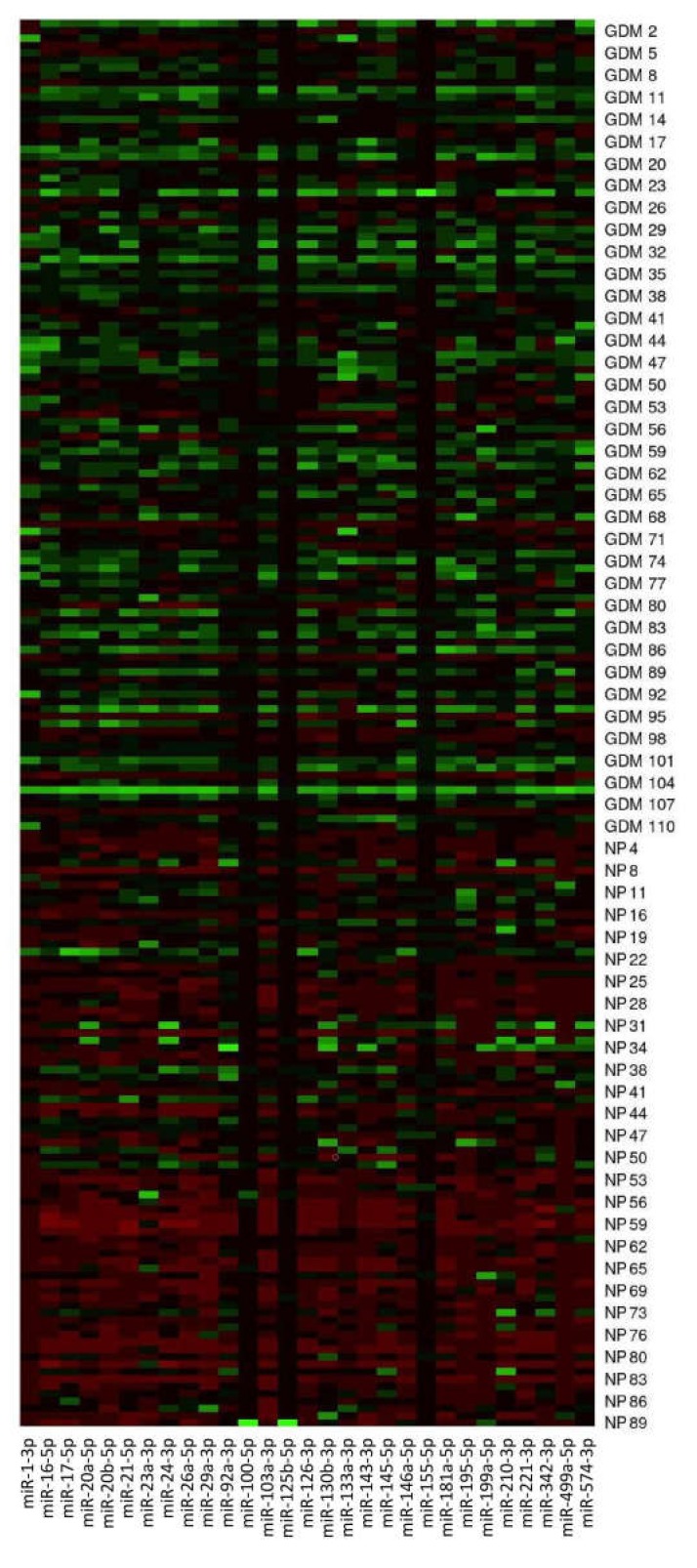
Postpartal microRNA expression profile in mothers with a history of GDM pregnancies. MicroRNA gene expression data (2^−^^ΔΔCt^) are visualized using the heat map. In this setting, each row represents a sample (GDM1–GDM111, NP1–NP89) and each column represents a microRNA gene. The color and intensity of the boxes are used to represent changes of gene expression (2^−^^ΔΔCt^). Green color indicates upregulation, and red color indicates downregulation. NP, normal pregnancies.

**Figure 2 ijms-21-02437-f002:**
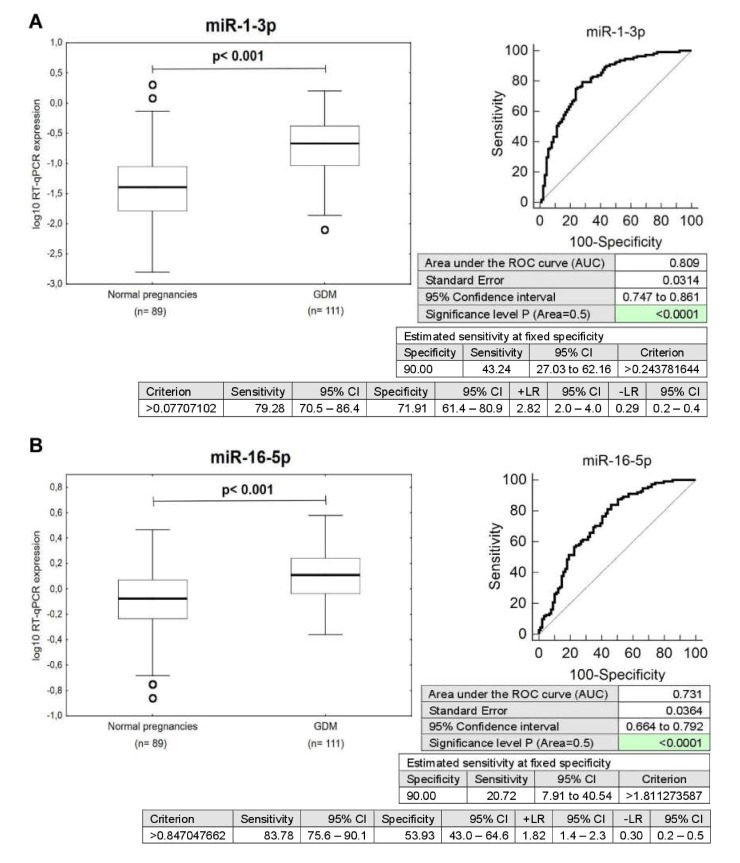
Postpartal microRNA expression profile in mothers with a history of GDM pregnancies. (**A**–**Z**) Upregulation of miR-1-3p, miR-16-5p, miR-17-5p, miR-20a-5p, miR-20b-5p, miR-21-5p, miR-23a-3p, miR-24-3p, miR-26a-5p, miR-29a-3p, miR-100-5p, miR-103a-3p, miR-125b-5p, miR-126-3p, miR-130b-3p, miR-133a-3p, miR-143-3p, miR-145-5p, miR-146a-5p, miR-181a-5p, miR-195-5p, miR-199a-5p, miR-221-3p, miR-342-3p, miR-499a-5p, and miR-574-3p was observed in mothers after GDM pregnancies when the comparison to the controls was performed using Mann–Whitney test. Receivers operating characteristic (ROC) curves were constructed to calculate the area under the curve (AUC), the best cut-off point (criterion), the sensitivity, specificity, likelihood ratio positive (LR+), and likelihood ratio negative (LR−) for particular microRNA. In addition, respective sensitivity at 90.0% specificity was reported for miR-1-3p (43.24%), miR-16-5p (20.72%), miR-17-5p (21.62%), miR-20a-5p (10.81%), miR-20b-5p (38.74%), miR-21-5p (28.83%), miR-23a-3p (17.12%), miR-24-3p (0.90%), miR-26a-5p (20.72%), miR-29a-3p (21.62%), miR-100-5p (13.51%), miR-103a-3p (30.63%), miR-125b-5p (9.91%), miR-126-3p (9.91%), miR-130b-3p (3.60%), miR-133a-3p (18.92%), miR-143-3p (13.51%), miR-145-5p (7.21%), miR-146a-5p (17.12%), miR-181a-5p (31.53%), miR-195-5p (16.22%), miR-199a-5p (15.32%), miR-221-3p (31.53%), miR-342-3p (1.8%), miR-499a-5p (28.83%), and miR-574-3p (10.81%).

**Figure 3 ijms-21-02437-f003:**
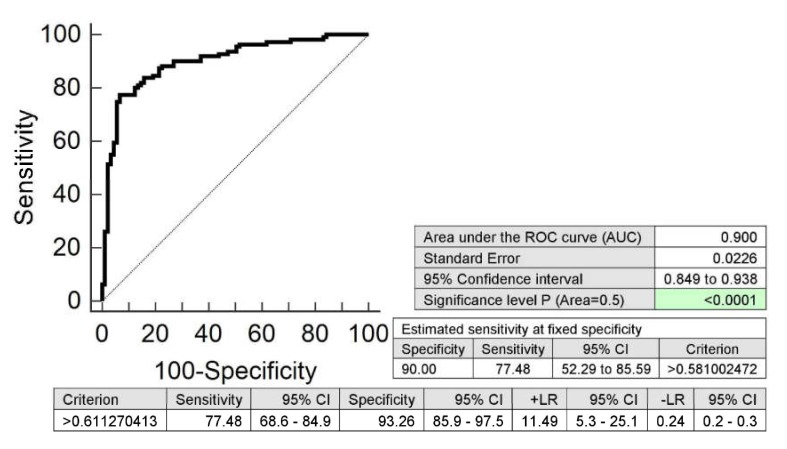
Combined postpartal screening of microRNAs in the identification of mothers with a history of GDM at an increased cardiovascular risk. Postpartal combined screening of miR-1-3p, miR-16-5p, miR-17-5p, miR-20b-5p, miR-21-5p, miR-23a-3p, miR-26a-5p, miR-29a-3p, miR-103a-3p, miR-133a-3p, miR-146a-5p, miR-181a-5p, miR-195-5p, miR-199a-5p, miR-221-3p, and miR-499a-5p showed the highest accuracy for the identification of mothers with a prior exposure to GDM at a higher risk of later development of diabetes mellitus, and cardiovascular/cerebrovascular diseases.

**Figure 4 ijms-21-02437-f004:**
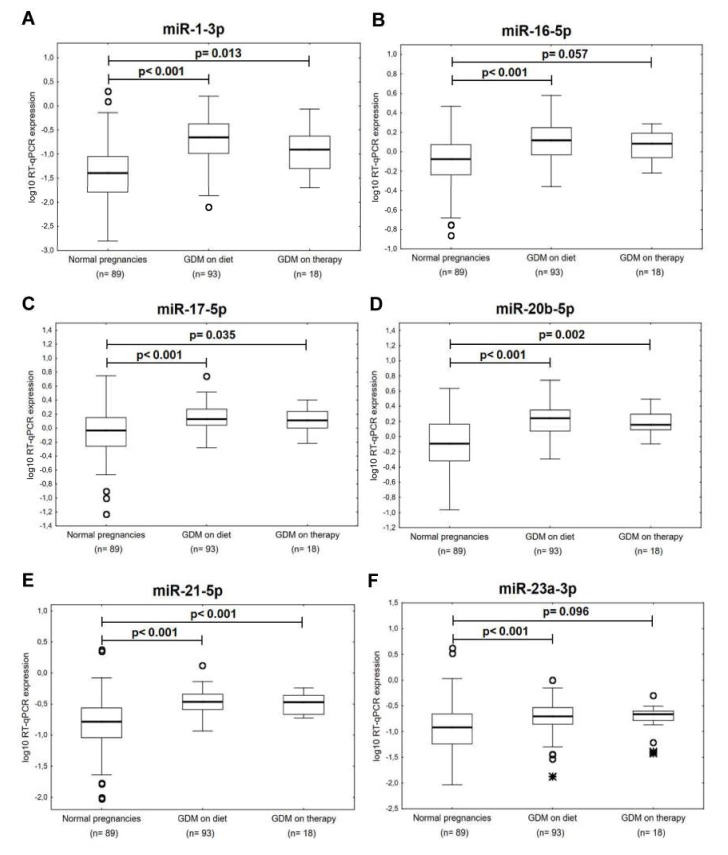
Postpartal microRNA expression profile in mothers with a history of GDM pregnancies with regard to the treatment strategies. (**A**–**O**) Upregulation or trend towards upregulation of miR-1-3p, miR-16-5p, miR-17-5p, miR-20b-5p, miR-21-5p, miR-23a-3p, miR-26a-5p, miR-29a-3p, miR-103a-3p, miR-146a-5p, miR-181a-5p, miR-195-5p, miR-199a-5p, miR-221-3p, and miR-499a-5p was observed in mothers after GDM pregnancies regardless of the treatment strategies (diet only and/or the combination of diet and therapy). (**P**) Upregulation of miR-133a-3p was observed in mothers after GDM pregnancies on diet only.

**Table 1 ijms-21-02437-t001:** The role of differentially expressed microRNAs in mothers with a history of GDM in the pathogenesis of diabetes mellitus, cardiovascular/cerebrovascular diseases.

miRBase ID	Gene Location on Chromosome	Role in the Pathogenesis of Cardiovascular/Cerebrovascular Diseases
hsa-miR-1-3p	20q13.3 18q11.2 [11]	Acute myocardial infarction, heart ischemia, post-myocardial infarction complications [12], diabetes mellitus [13,14], vascular endothelial dysfunction [15]
hsa-miR-16-5p	13q14.2	Myocardial infarction [16,17], heart failure [18], acute coronary syndrome, cerebral ischaemic events [19], gestational diabetes mellitus [20,21], diabetes mellitus [22,23,24]
hsa-miR-17-5p	13q31.3 [25,26]	Cardiac development [27], ischemia/reperfusion-induced cardiac injury [28], kidney ischemia-reperfusion injury [29], diffuse myocardial fibrosis in hypertrophic cardiomyopathy [30], acute ischemic stroke [31], coronary artery disease [32], adipogenic differentiation [33], gestational diabetes mellitus [20,21], diabetes mellitus [24,34]
hsa-miR-20a-5p	13q31.3 [35]	Pulmonary hypertension [36], gestational diabetes mellitus [20,21,37], diabetic retinopathy [38], diabetes with abdominal aortic aneurysm [39]
hsa-miR-20b-5p	Xq26.2 [35]	Hypertension-induced heart failure [40], insulin resistance [41], T2DM [42,43], diabetic retinopathy [44]
hsa-miR-21-5p	17q23.2 [45]	Homeostasis of the cardiovascular system [46], cardiac fibrosis and heart failure [47,48], ascending aortic aneurysm [49], regulation of hypertension-related genes [50], myocardial infarction [51], insulin resistance [41], T2DM [52], T2DM with major cardiovascular events [53], T1DM [54,55,56], diabetic nephropathy [57]
hsa-miR-23a-3p	19p13.12	Heart failure [58], coronary artery disease [59], cerebral ischemia-reperfusion [60], vascular endothelial dysfunction [15], small and large abdominal aortic aneurysm [61], obesity and insulin resistance [62]
hsa-miR-24-3p	19p13.12	Asymptomatic carotid stenosis [63], familial hypercholesterolemia and coronary artery disease [64], angina pectoris [65], ischemic dilated cardiomyopathy [66], small and large abdominal aortic aneurysm [61], myocardial ischemia/reperfusion [67,68], diabetes mellitus [14,24,52,54]
hsa-miR-26a-5p	3p22.2 12q14.1 [69]	Heart failure, cardiac hypertrophy [70], myocardial infarction [51,71,72], ischemia/reperfusion injury [73], pulmonary arterial hypertension [74], T1DM [75], diabetic nephropathy [57]
hsa-miR-29a-3p	7q32.3	Ischemia/reperfusion-induced cardiac injury [76], cardiac cachexia, heart failure [77], atrial fibrillation [78], diffuse myocardial fibrosis in hypertrophic cardiomyopathy [30], coronary artery disease [79], pulmonary arterial hypertension [74], gestational diabetes mellitus [80], diabetes mellitus [13,23,81,82]
hsa-miR-100-5p	11q24.1	Failing human heart, idiopathic dilated cardiomyopathy, ischemic cardiomyopathy [66], regulation of hypertension-related genes [50], T1DM [54]
hsa-miR-103a-3p	5q34 20p13 [83]	Hypertension [84], hypoxia-induced pulmonary hypertension [85], myocardial ischemia/reperfusion injury, acute myocardial infarction [84], ischemic dilated cardiomyopathy [66], obesity, regulation of insulin sensitivity [86], T1DM [87]
hsa-miR-125b-5p	11q24.1 21q21.1 [88]	Acute ischemic stroke [89], acute myocardial infarction [90,91], ischemic dilated cardiomyopathy [66], ascending aortic aneurysm [49], gestational diabetes mellitus [92], T1DM [93,94], T2DM [95]
hsa-miR-126-3p	9q34.3 [96]	Acute myocardial infarction [72], T2DM [53,97], T2DM with major cardiovascular events [53], gestational diabetes mellitus [98]
hsa-miR-130b-3p	22q11.21	Hypertriglyceridemia [99,100], intracranial aneurysms [101], hyperacute cerebral infarction [102], T2DM [52,103,104], gestational diabetes mellitus [98]
hsa-miR-133a-3p	18q11.2 20q13.33 [105]	Heart failure [106], myocardial fibrosis in hypertrophic cardiomyopathy [30,107], arrhythmogenesis in the hypertrophic and failing hearts [108,109], coronary artery calcification [110], ascending aortic aneurysm [49], diabetes mellitus [13,14]
hsa-miR-143-3p	5q33	Intracranial aneurysms [111], coronary heart disease [112], myocardial infarction [113], myocardial hypertrophy [114], dilated cardiomyopathy [115], pulmonary arterial hypertension [116], acute ischemic stroke [89], ascending aortic aneurysm [49]
hsa-miR-145-5p	5q33	Hypertension [117,118], dilated cardiomyopathy [119], myocardial infarction [120,121], stroke [121], acute cerebral ischemic/reperfusion [122], T2DM [24,123], T1DM [52], diabetic retinopathy [124], gestational diabetes mellitus [125]
hsa-miR-146a-5p	5q33.3 [126,127]	Angiogenesis [128], hypoxia, ischemia/reperfusion-induced cardiac injury [129], myocardial infarction [17], coronary atherosclerosis, coronary heart disease in patients with subclinical hypothyroidism [130], acute ischemic stroke, acute cerebral ischemia [131], T2DM [24,52], T1DM [75], diabetic nephropathy [57]
hsa-miR-181a-5p	1q32.1 9q33.3 [132]	Regulation of hypertension-related genes [50], atherosclerosis [132], metabolic syndrome, coronary artery disease [133], non-alcoholic fatty liver disease [134], ischaemic stroke, transient ischaemic attack, acute myocardial infarction [135,136], obesity and insulin resistance [62,132,133], T1DM [52,137], T2DM [132,136]
hsa-miR-195-5p	17p13.1 [138]	Cardiac hypertrophy, heart failure [139,140], abdominal aortic aneurysms [141], aortic stenosis [142], T2DM [123], gestational diabetes mellitus [143]
hsa-miR-199a-5p	1q24.3 19p13.2	Hypertension [144], congenital heart disease [145], pulmonary artery hypertension [146], unstable angina [147], diabetic retinopathy [148], T1DM, T2DM, gestational diabetes mellitus [149]
hsa-miR-221-3p	Xp11.3	Asymptomatic carotid stenosis [63], cardiac amyloidosis [150], heart failure [151], atherosclerosis [152,153], aortic stenosis [154], acute myocardial infarction [155], acute ischemic stroke [156], focal cerebral ischemia [157], pulmonary artery hypertension [158], obesity [159]
hsa-miR-342-3p	14q32.2	Cardiac amyloidosis [150], obesity [160], T1DM [52,149,161], T2DM [149,162,163], GDM [149], endothelial dysfunction [164]
hsa-miR-499a-5p	20q11.22	Myocardial infarction [17,165], hypoxia [166], cardiac regeneration [167], vascular endothelial dysfunction [15]
hsa-miR-574-3p	4p14	Myocardial infarction [168], coronary artery disease [100], cardiac amyloidosis [150], stroke [169], T2DM [104,170]

**Table 2 ijms-21-02437-t002:** A list of predicted targets of appropriate microRNAs dysregulated in whole peripheral blood of patients with a history of GDM in relation to insulin signaling pathway using miRWalk2.0 database.

microRNA	Predicted Targets
miR-1	CALM2, CBL, IKBKB, KRAS, PHKG2, PIK3R5, PTPN1, PTPRF, TRIP10
miR-16-5p	IKBKB, PHKA1, PRKAR1A, MAP2K1, RAF1, IRS4, MKNK1, EXOC7, FASN
miR-17-5p	PHKA2, CRK, GRB2, PDE3A, PHKG1, PIK3R2, PRKAA2, PRKAR2A, MAPK9, PRKX, PPP1R3B, HK1, PCK1, SREBF1
miR-20a-5p	BRAF, MKNK2, CRK, SLC2A4, TRIP10, KRAS, PCK1
miR-20b-5p	CRK, GRB2, PDE3A, PHKA2, PHKG1, PIK3R2, PRKAA2, PRKAR2A, MAPK9, PRKX, PPP1R3B, HK1, PCK1, SREBF1
miR-21-5p	PPP1R3A, PPP1R3D
miR-23a-3p	G6PC, IRS2, IKBKB, PIK3CB, FASN, PRKAG3
miR-24-3p	IKBKB, PIK3CB, PTPRF, SHC2, INPP5K, PHKG1, PRKAG3
miR-26a-5p	G6PC, MKNK2, GYS2, PPP1R3D, RHOQ, KRAS, PRKAG1, PYGL
miR-29a-3p	NRAS, EIF4E2, CALM3
miR-100-5p	MTOR
miR-103a-3p	FAS, RAPGEF1, PDE3B, ACACB, PHKAR1A, PRKCI, IRS2, LIPE, PRKC2, MAPK3, TRIP10, CBLC, CALML5
miR-125b-5p	PHKA1, RAF1, ACACB, FLOT2, HK2, EIF4E2, PHKG1
miR-126-3p	TSC1
miR-130b-3p	RPS6KB1, MAP2K1, SOS2, FLOT2, EXOC7, PHKG2, PIK3CA, PRKC2, TSC2, PRKAG3
miR-133a-3p	PRKAB1
miR-143-3p	FOXO1, KRAS, HK2, PHKG2, MAPK3, MAPK9, SREBF1
miR-145-5p	IRS1, IRS2, PIK3R5, PRKAG3
miR-181a-5p	NRAS, AKT3, SOCS4, HK2, PDE3B, PPP1R3C, PRKAR2A, MAPK1, PPP1R3D, PRKAA1
miR-195-5p	IKBKB, PHKA1, PRKAR1A, MAP2K1, RAF1, IRS4, MKNK1, EXOC7, FASN
miR-199a-5p	PRKX, PCK1, IRS1, SLC2A4, MAPK9, RHEB, PRKAR1A
miR-221-3p	AKT3, PIK3CD, MAPK10
miR-342-3p	PDPK1, INSR, PHKG2, EIF4E2, PIK3CD, RPS6KB2
miR-499a-5p	AKT2, CRK, KRAS, PIK3CD, PRKAR1A, SOS2
miR-574-3p	PRKCZ, HK1

**Table 3 ijms-21-02437-t003:** A list of predicted targets of appropriate microRNAs dysregulated in whole peripheral blood of patients with a history of GDM in relation to type 1 diabetes mellitus pathway using miRWalk2.0 database.

microRNA	Predicted Targets
miR-1	CD28, LTA
miR-16-5p	HLA-DQA1
miR-17-5p	FASLG, CD28, HLA-DQA, GAD2, HLA-DPA1
miR-20a-5p	IL12A
miR-20b-5p	HLA-DOA, FASLG, CD28, HLA-DPA1, GAD2
miR-21-5p	HLA-DPB1, FASLG, IL12A
miR-23a-3p	IFNG
miR-24-3p	CD28, CD86, IFNG, FASLG, IL1B, HLA-DOA
miR-26a-5p	HLA-DPB1, HLA-DPA1, HLA-A, IFNG
miR-29a-3p	HLA-DQA2
miR-103a-3p	HLA-DPB1, CD80
miR-125b-5p	PRF-1
miR-130b-3p	HLA-DOA, HLA-DQB1, HLA-A, HLA-B, HLA-C, HLA-G
miR-133a-3p	HLA-DOA, CD28, GAD2, LTA
miR-143-3p	HLA-DOA, HLA-DPB1, HLA-DPA1, IFNG, CD28
miR-146a-5p	CD80, CD86, PRF1, ICA1, HLA-C, GAD2
miR-181a-5p	IL2, HLA-E, IL1A
miR-195-5p	HLA-DQA1
miR-199a-5p	G2MB, ICA1,TNF
miR-221-3p	HLA-DQA1, PTPRN
miR-342-3p	PTPRN2, HLA-A, HLA-F

**Table 4 ijms-21-02437-t004:** A list of predicted targets of appropriate microRNAs dysregulated in whole peripheral blood of patients with a history of GDM in relation to type 2 diabetes mellitus pathway using miRWalk2.0 database.

microRNA	Predicted Targets
miR-16-5p	CACNA1E, IKBKB, IRS4
miR-17-5p	PIK3R2, MAPK9, HK1
miR-20a-5p	SLC2A4
miR-20b-5p	MAPK9, HK1
miR-23a-3p	IRS2, IKBKB, PIK3CB
miR-24-3p	IKBKB, PIK3CB, KCNJ11
miR-26a-5p	PRKCD
miR-29a-3p	CACNA1A, CACNA1B
miR-100-5p	MTOR
miR-103a-3p	CACNA1E, IRS2, PRKCZ, MAPK3
miR-125b-5p	HK2
miR-130b-3p	PIK3CA, PRKCZ
miR-143-3p	CACNA1A, HK2, PRKCE, MAPK3, MAPK9
miR-145-5p	IRS2, IRS1, PIK3R5
miR-146a-5p	PRKCE
miR-181a-5p	SOCS4, HK2, MAPK1
miR-195-5p	CACNA1E, IKBKB, IRS4
miR-199a-5p	IRS1, SLC2A4, MAPK9, PKM, TNF, CACNA1G
miR-221-3p	PIK3CD, MAPK10
miR-342-3p	CACNA1C, INSR, PIK3CD
miR-499a-5p	PRKCE, PIK3CD
miR-574-3p	PRKCZ, HK1

The miRWalk database and the predicted target module were used to provide information on predicted interactions between appropriate microRNAs and specific genes involved in human biological pathways such as insulin signaling, type 1 diabetes mellitus, and type 2 diabetes mellitus.

**Table 5 ijms-21-02437-t005:** Characteristics of cases and controls, pre-existing cardiovascular risk factors before gestation.

	Normotensive Term Pregnancies (*n* = 89)	GDM on Diet Only (*n* = 93)	GDM on Diet and Therapy (*n* = 18)
Rheumatoid arthritis	0 (0%)	1 (1.08%)	0 (0%)
SLE	0 (0%)	0 (0%)	0 (0%)
On blood pressure treatment	0 (0%)	0 (0%)	0 (0%)
Hypercholesterolemia	0 (0%)	1 (1.08%)	0 (0%)
Dispensarisation at Dpt. of Cardiology (valve problems and heart defects)	0 (0%)	4 (4.30%) Mitral valve prolapse Heart arrhythmia Hypertrophic cardiomyopathy AV nodal reentrant tachycardia	1 (5.55%) Mitral valve regurgitation
Chronic venous insufficiency	0 (0%)	0 (0%)	0 (0%)
Thrombosis	0 (0%)	2 (2.15%)	0 (0%)
Presence of risk factors for chronic kidney disease	0 (%)	6 (6.45%) Pyelonephritis Glomerulonephritis Ureteral stent Nephrolithiasis Hydronephrosis	0 (0%)
Chronic kidney disease	0 (%)	1 (0.98%) Chronic renal insufficiency	0 (0%)

**Table 6 ijms-21-02437-t006:** Characteristics of cases and controls, data gained at follow-up.

	Normotensive Term Pregnancies (*n* = 89)	GDM on Diet Only (*n* = 93)	GDM on Diet and Therapy (*n* = 18)	*p*-value^1^	*p*-value^2^
Age (years)	38.33 ± 0.38	38.70 ± 0.37	38.61 ± 0.80	0.465	0.564
Time elapsed since delivery (years)	5.75 ± 0.20	5.49 ± 0.12	5.33 ± 0.20	0.965	0.045
BMI	23.15 ± 0.38	23.85 ± 0.37	27.21 ± 0.83	1.000	1.000
Normal (< 25)	67 (75.28%)	70 (75.27%)	5 (27.78%)	0.633	< 0.001
Overweight (≥ 25 <30)	18 (20.22%)	16 (17.20%)	9 (50.00%)		
Obese (≥ 30)	4 (4.49%)	7 (7.53%)	4 (22.22%)		
Smoking				-	-
Non-Smoker	54 (60.67%)	78 (83.87%)	16 (88.88%)		
Ex-smoker	21 (23.60%)	6 (6.45%)	2 (11.11%)		
Smoker	14 (15.73%)	9 (9.68%)	0 (0%)		
Angina or heart attack in a first degree relative before the age of 60 years	2 (2.22%)	6 (6.45%)	2 (11.11%)	-	-
Atrial fibrillation	0 (0%)	0 (0%)	0 (0%)	-	-
DM type I	0 (0%)	0 (0%)	0 (0%)	-	-
DM type II	0 (0%)	0 (0%)	0 (0%)	-	-
Fasting serum glucose levels				-	-
Normal (3.33–5.59 mmol/L)	89 (100.0%)	90 (96.77%)	18 (100.0%)		
High (> 5.59 mmol/L)	0 (0%)	3 (3.23%)	0 (0%)		
Fasting serum total cholesterol levels				-	-
Normal (2.9–5.0 mmol/L)	43 (48.31%)	51 (54.84%)	10 (55.55%)		
High (> 5.0–7.9 mmol/L)	46 (51.69%)	41(44.09%)	8 (44.44%)		
Critical (≥ 8.0 mmol/L)	0 (0%)	1 (1.08%)	0 (0%)		
Fating serum LDL cholesterol levels				0.987	0.281
Normal (1.2–3.0 mmol/L)	42 (47.19%)	44 (47.31%)	6 (33.33%)		
High (> 3.0 mmol/L)	47 (52.81%)	49 (52.69%)	12 (66.67%)		
SBP				-	-
Normal (< 140 mmHg)	89 (100.0%)	92 (98.92%)	18 (100.0%)		
High (≥ 140–179 mmHg)	0 (0%)	1 (1.08%)	0 (0%)		
Critical (≥ 180 mmHg)	0 (0%)	0 (0%)	0 (0%)		
DBP				-	-
Normal (< 90 mmHg)	88 (98.88%)	90 (96.77%)	17 (94.44%)		
High (≥ 90–109 mmHg)	1 (1.12%)	3 (3.23%)	1 (5.56%)		
Critical (≥ 110 mmHg)	0 (0%)	0 (0%)	0 (0%)		
On blood pressure treatment	0 (0%)	0 (0%)	0 (0%)	-	-
Chronic kidney disease	0 (%)	1 (0.98%) Chronic renal insufficiency	0 (0%)	-	-
Chronic venous insufficiency	0 (0%)	0 (0%)	0 (0%)	-	-
Thrombosis	0 (0%)	2 (2.15%)	0 (0%)	-	-
Relative QRISK^®^3-2018 risk score	0.926 ± 0.05	0.889 ± 0.04	1.022 ± 0.08	1.0	1.0
Hormonal contraceptive use				0.040	0.474
No	12 (13.48%)	5 (5.38%)	2 (11.11%)		
In the past	63 (70.79%)	80 (86.02%)	15 (83.33%)		
Yes	14 (15.73%)	8 (8.60%)	1 (5.56%)		
Total number of pregnancies per patient				0.924	0.440
1	9 (10.11%)	8 (8.60%)	2 (11.11%)		
2	44 (49.44%)	48 (51.61%)	6 (33.33%)		
3+	36 (40.45%)	37 (39.78%)	10 (55.55%)		
Total parity per patient				0.991	0.217
1	13 (14.61%)	13 (13.98%)	2 (11.11%)		
2	62 (69.66%)	65 (69.89%)	10 (55.55%)		
3+	14 (15.73%)	15 (16.13%)	6 (33.33%)		

Data are presented as mean ± SE for continuous variables and as number (percent) for categorical variables. Statistically significant results are marked in bold. Continuous variables were compared using Mann–Whitney test. *p*-value^1^, the comparison among normal pregnancies and GDM on diet; *p*-value^2^, the comparison among normal pregnancies and GDM on diet and therapy, respectively. Categorical variables were compared using a chi-square test. GA, gestational age; SBP, systolic blood pressure; DBP, diastolic blood pressure; CS, caesarean section.

**Table 7 ijms-21-02437-t007:** Characteristics of cases and controls, data gained during gestation.

	Normotensive Term Pregnancies (*n* = 89)	GDM on Diet Only (*n* = 93)	GDM on Diet and Therapy (*n* = 18)	*p*-value^1^	*p*-value^2^
Maternal age at delivery (years)	32.62 ± 0.36	33.20 ± 0.35	33.28 ± 0.83	0.111	1.000
GA at delivery (weeks)	39.90 ± 0.10	39.59 ± 0.09	39.20 ± 0.23	0.961	1.000
Fetal birth weight (g)	3397.53 ± 40.42	3436.56 ± 35.96	3526.11 ± 63.35	1.000	1.000
Mode of delivery				< 0.001	0.015
Vaginal	82 (92.13%)	57 (61.29%)	13 (72.22%)		
CS	7 (7.87%)	36 (38.71%)	5 (27.78%)		
Fetal sex				0.476	0.281
Boy	47 (52.81%)	54 (58.06%)	12 (66.67%)		
Girl	42 (47.19%)	39 (41.94%)	6 (33.33%)		
Infertility treatment				0.006	0.846
Yes	4 (4.49%)	16 (17.20%)	1 (5.56%)		
No	85 (95.51%)	77 (82.80%)	17 (94.44%)		

Data are presented as mean ± SE for continuous variables and as number (percent) for categorical variables. Statistically significant results are marked in bold. Continuous variables were compared using Mann–Whitney test. *p*-value^1^, the comparison among normal pregnancies and GDM on diet; *p*-value^2^, the comparison among normal pregnancies and GDM on diet and therapy, respectively. Categorical variables were compared using a chi-square test. GA, gestational age; SBP, systolic blood pressure; DBP, diastolic blood pressure; CS, caesarean section.

**Table 8 ijms-21-02437-t008:** Characteristics of microRNAs involved in the study.

Assay Name	miRBase ID	NCBI Location Chromosome	microRNA Sequence
hsa-miR-1	hsa-miR-1-3p	Chr20: 61151513-61151583 [+]	5´-UGGAAUGUAAAGAAGUAUGUAU-3´
hsa-miR-16	hsa-miR-16-5p	Chr13: 50623109-50623197 [−]	5´-UAGCAGCACGUAAAUAUUGGCG- 3´
hsa-miR-17	hsa-miR-17-5p	Chr13: 92002859-92002942 [+]	5´-CAAAGUGCUUACAGUGCAGGUAG-3´
hsa-miR-20a	hsa-miR-20a-5p	Chr13: 92003319-92003389 [+]	5´-UAAAGUGCUUAUAGUGCAGGUAG-3´
hsa-miR-20b	hsa-miR-20b-5p	ChrX: 133303839-133303907 [−]	5´-CAAAGUGCUCAUAGUGCAGGUAG-3´
hsa-miR-21	hsa-miR-21-5p	Chr17: 57918627-57918698 [+]	5´-UAGCUUAUCAGACUGAUGUUGA-3´
hsa-miR-23a	hsa-miR-23a-3p	Chr19: 13947401-13947473 [−]	5´-AUCACAUUGCCAGGGAUUUCC-3´
hsa-miR-24	hsa-miR-24-3p	Chr19: 13947101-13947173 [−]	5´-UGGCUCAGUUCAGCAGGAACAG-3´
hsa-miR-26a	hsa-miR-26a-5p	Chr3: 38010895-38010971 [+]	5´-UUCAAGUAAUCCAGGAUAGGCU-3´
hsa-miR-29a	hsa-miR-29a-3p	Chr7: 130561506-130561569 [−]	5´-UAGCACCAUCUGAAAUCGGUUA-3´
hsa-miR-92a	hsa-miR-92a-3p	Chr13: 92003568-92003645 [+]	5´-UAUUGCACUUGUCCCGGCCUGU-3´
hsa-miR-100	hsa-miR-100-5p	Chr11: 122022937-122023016 [−]	5´-AACCCGUAGAUCCGAACUUGUG-3´
hsa-miR-103	hsa-miR-103a-3p	Chr20: 3898141-3898218 [+]	5´-AGCAGCAUUGUACAGGGCUAUGA-3´
hsa-miR-125b	hsa-miR-125b-5p	Chr21: 17962557-17962645 [+]	5´-UCCCUGAGACCCUAACUUGUGA-3´
hsa-miR-126	hsa-miR-126-3p	Chr9: 139565054-139565138 [+]	5´-UCGUACCGUGAGUAAUAAUGCG-3´
hsa-miR-130b	hsa-miR-130b-3p	Chr22: 22007593-22007674 [+]	5´-CAGUGCAAUGAUGAAAGGGCAU-3´
hsa-miR-133a	hsa-miR-133a-3p	Chr20: 61162119-61162220 [+]	5´-UUUGGUCCCCUUCAACCAGCUG-3´
hsa-miR-143	hsa-miR-143-3p	Chr5: 148808481-148808586 [+]	5´-UGAGAUGAAGCACUGUAGCUC-3´
hsa-miR-145	hsa-miR-145-5p	Chr5: 148810209-148810296 [+]	5´-GUCCAGUUUUCCCAGGAAUCCCU-3´
hsa-miR-146a	hsa-miR-146a-5p	Chr5: 159912359-159912457 [+]	5´-UGAGAACUGAAUUCCAUGGGUU-3´
hsa-miR-155	hsa-miR-155-5p	Chr21: 26946292-26946356 [+]	5´-UUAAUGCUAAUCGUGAUAGGGGU-3´
hsa-miR-181a	hsa-miR-181a-5p	Chr9: 127454721-127454830 [+]	5´-AACAUUCAACGCUGUCGGUGAGU-3´
hsa-miR-195	hsa-miR-195-5p	Chr17: 6920934-6921020 [−]	5´-UAGCAGCACAGAAAUAUUGGC-3´
hsa-miR-199a	hsa-miR-199a-5p	Chr19: 10928102-10928172 [−]	5´-CCCAGUGUUCAGACUACCUGUUC-3´
hsa-miR-210	hsa-miR-210-3p	Chr11: 568089-568198 [−]	5´-CUGUGCGUGUGACAGCGGCUGA-3´
hsa-miR-221	hsa-miR-221-3p	ChrX: 45605585-45605694 [−]	5´-AGCUACAUUGUCUGCUGGGUUUC-3´
hsa-miR-342-3p	hsa-miR-342-3p	Chr14: 100575992-100576090 [+]	5´-UCUCACACAGAAAUCGCACCCGU-3´
mmu-miR-499	hsa-miR-499a-5p	Chr20: 33578179-33578300 [+]	5´-UUAAGACUUGCAGUGAUGUUU-3´
hsa-miR-574-3p	hsa-miR-574-3p	Chr4: 38869653-38869748 [+]	5´-CACGCUCAUGCACACACCCACA-3´

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
