# Peer review of "Diabetes Mellitus and Cardiovascular Risk Assessment in Mothers with a History of Gestational Diabetes Mellitus Based on Postpartal Expression Profile of MicroRNAs Associated with Diabetes Mellitus and Cardiovascular and Cerebrovascular Diseases"

_ijms, 2020, doi:10.3390/ijms21072437_

Round 1

Reviewer 1 Report

The authors submitted their manuscript to the IJSM with the above title. The topic is interesting and it is in the focus of interest of other researchers and clinicians.

They involved 111 patients with GDM (93 on diet and 11 under treatment) and 89 healthy control in the study..

Ethical permission was obtained and patients and controls agreed in their involvement in the study.

The performed molecular genetic techniques and statistical analysis are adquate and introduced in details..

We can see huge amount of data in the Results section they are shown in tables and figures .

Epigenetic changes were observed for diabetes mellitus and cardiovascvular/cerebrovascular diseases in the peripherial blood isolates. The authors observed several upregulated miRNAs in the samples. There were nine upregulated miRNAs which are unique for development of GDM. May be they could serve as biomarkers later. It seems these miRNAs have longer effect on the epigenome

The ms is clearly written and we can follow up. 

Author Response

Response to Reviewer 1 Comments

Point 1:

The authors submitted their manuscript to the IJSM with the above title. The topic is interesting and it is in the focus of interest of other researchers and clinicians.

They involved 111 patients with GDM (93 on diet and 11 under treatment) and 89 healthy control in the study..

Ethical permission was obtained and patients and controls agreed in their involvement in the study.

The performed molecular genetic techniques and statistical analysis are adquate and introduced in details..

We can see huge amount of data in the Results section they are shown in tables and figures .

Epigenetic changes were observed for diabetes mellitus and cardiovascvular/cerebrovascular diseases in the peripherial blood isolates. The authors observed several upregulated miRNAs in the samples. There were nine upregulated miRNAs which are unique for development of GDM. May be they could serve as biomarkers later. It seems these miRNAs have longer effect on the epigenome

The ms is clearly written and we can follow up.

Response 1: We thank to the reviewer for positive evaluation. No questions to be answered were raised.

Reviewer 2 Report

Ilona Hromadnikova et al showed a comparative analysis of the expression of selected microRNA in whole blood from gestational diabetes mellitus mothers (GDM). Although the study is original and novel, the results are poor. Authors only showed differential expression in a group of miRNAs that they selected based on literature. 

The first issue is that they did not performed the whole profile of microRNA (more than 2000) which would allow the identification of better candidates. 

Moreover the only relevant results of the whole paper is a list of selected miRNA deregulated. The Figure 3 showed differences in relation to control but no differences when GDM on diet and therapy are compared.

The way to represent the result is weak. They must provide a whole figure in which they can summarize the results (heat maps, tables etc).

The title of the results are not appropriate for a scientific journal giving a large list of miRNAs.

Table 1 and 2 should be included in supplemental material since they are not relevant as results or methods.

Authors did not show the relationship of the expression level of miRNAs with clinical variables (correlation and association studies) and with potential targets that can explain the impact of the deregulation of these microRNAs. A bioinformatic approach to detect potential common mRNA targets in the context of diabetes must be included (TargetScan , miRANDA etc).

Analysing a list of selected miRNAs is not enough for a relevant scientific journal. I would aim authors to check other studies using microRNAs as biomarkers to improve the quality of the results as well the presentation of the data.

Author Response

Response to Reviewer 2 Comments

Point 1: Ilona Hromadnikova et al showed a comparative analysis of the expression of selected microRNA in whole blood from gestational diabetes mellitus mothers (GDM). Although the study is original and novel, the results are poor. Authors only showed differential expression in a group of miRNAs that they selected based on literature. The first issue is that they did not performed the whole profile of microRNA (more than 2000) which would allow the identification of better candidates. 

Response 1: Thank you for this comment. The aim of our study was to compare microRNA expression profile of patients with a history of GDM to those ones with a history of preeclampsia, fetal growth restriction or gestational hypertension. Since we have already published microRNA expression profiles of patients with a history of preeclampsia, FGR and/or gestational hypertension (Int J Cardiol. 2019 Sep 15;291:158-167), we were interested how the profile of the same microRNAs will differ in patients with a history of GDM. That is the reason, why we did not perform the whole profiling of all human microRNAs in this particular setting. This study showed that patients with a history of GDM had much worse epigenetic profile than patients with a history of gestational hypertension, preeclampsia or fetal growth restriction (please see details in original version of the manuscript in discussion and conclusion sections).

Point 2: Moreover the only relevant results of the whole paper is a list of selected miRNA deregulated. The Figure 3 showed differences in relation to control but no differences when GDM on diet and therapy are compared.

Response 2: As suggested this information was added. It was not commented in original version of manuscript, since no statistical difference between these two groups of patients was observed.

2.2. Expression profile of microRNAs associated with diabetes mellitus and cardiovascular/cerebrovascular diseases in mothers after GDM pregnancies with regard to the treatment strategies (diet only and/or diet and therapy)

No difference in microRNA expression profiles was observed between the groups of mothers with a history of GDM on diet only and on the combination of diet and therapy (p=1.0 for nearly all examined microRNAs).

Point 3: The way to represent the result is weak. They must provide a whole figure in which they can summarize the results (heat maps, tables etc).

Response 3: As suggested this information was added.

Figure 1. Postpartal microRNA expression profile in mothers with a history of GDM pregnancies. MicroRNA gene expression data (2-DDCt) are visualised using the heat map. In this setting, each row represents a sample (GDM1 - GDM111, NP1 - NP89) and each column represents a microRNA gene. The colour and intensity of the boxes are used to represent changes of gene expression (2-DDCt). Green colour indicates up-regulation, and red colour indicates down-regulation.

Point 4: The title of the results are not appropriate for a scientific journal giving a large list of miRNAs.

Response 4: As suggested titles of Results subsections were changed.

2.1. Expression profile of microRNAs associated with diabetes mellitus and cardiovascular/cerebrovascular diseases in mothers after GDM pregnancies

2.2. Expression profile of microRNAs associated with diabetes mellitus and cardiovascular/cerebrovascular diseases in mothers after GDM pregnancies with regard to the treatment strategies (diet only and/or diet and therapy)

Point 5: Table 1 and 2 should be included in supplemental material since they are not relevant as results or methods.

Response 5: Table 1. summarizes the role of differentially expressed microRNAs in mothers with a history of GDM in the pathogenesis of diabetes mellitus, cardiovascular/cerebrovascular diseases. We feel that this is an important information highly related to the hypothesis of the study and therefore we would like to keep it in the main body of the manuscript. Table 2 contains characteristics of patients and controls, which we collected for the purpose of the study. We feel that this is an important information highly related to the results of the study and therefore we would like to keep it in the main body of the manuscript.

Point 6: Authors did not show the relationship of the expression level of miRNAs with clinical variables (correlation and association studies) and with potential targets that can explain the impact of the deregulation of these microRNAs. A bioinformatic approach to detect potential common mRNA targets in the context of diabetes must be included (TargetScan , miRANDA etc).

Analysing a list of selected miRNAs is not enough for a relevant scientific journal. I would aim authors to check other studies using microRNAs as biomarkers to improve the quality of the results as well the presentation of the data.

Response 6: Thank you for this comment. Association studies are very important and are part of our patent application (PCT), but there are not a part of this particular study, which involves just 111 patients with a history of GDM and 89 patients with normally ongoing gestation. To perform association studies and identify association between microRNA gene expression in whole peripheral blood and risky factors (hypertension, endothelial dysfunction, abnormal serum/plasma lipid profiles, etc.) sufficient amount of patients must be involved. Therefore for the purpose of our patent application we analysed altogether more than 500 patients. We made analysis on patients with a history of preeclampsia, gestational hypertension, fetal growth restriction, GDM and preterm birth. Significant correlations were identified. Nearly all dysregulated microRNAs were associated with any risk factor. Nevertheless, we still continue in collection of samples and data to get even higher number of patients on which association studies will be finally published. Our intention is to get as much convincing data as possible for our

potential investors.

Furthermore, we have already performed bioinformatics on some cardiovascular disease associated microRNAs when we studied their role in pathogenesis of pregnancy-related complications. Please see our published papers (Plos one 2015, Thrombosis Research 2016). So we did not consider this part crucial to be implemented in our original manuscript. Nevertheless, as required we implemented bioinformatics in a revised version of the manuscript. Please see results and methods sections.

Results

2.3. Information on microRNA-gene-biological pathway interactions

The extensive file of predicted targets of all microRNAs aberrantly expressed in whole peripheral blood of mothers with a history of GDM indicates that a large group of genes is involved in biological pathways related to insulin signaling, type 1 diabetes mellitus, and type 2 diabetes mellitus (Table 2).

Table 2a. A list of predicted targets of appropriate microRNAs dysregulated in whole peripheral blood of patients with a history of GDM in relation to insulin signaling pathway using miRWalk2.0 database

microRNA

Predicted targets

miR-1

CALM2, CBL, IKBKB, KRAS, PHKG2, PIK3R5, PTPN1, PTPRF, TRIP10

miR-16-5p

IKBKB, PHKA1, PRKAR1A, MAP2K1, RAF1, IRS4, MKNK1, EXOC7,FASN

miR-17-5p

PHKA2, CRK, GRB2, PDE3A, PHKG1, PIK3R2, PRKAA2, PRKAR2A, MAPK9, PRKX, PPP1R3B, HK1, PCK1, SREBF1

miR-20a-5p

BRAF, MKNK2, CRK, SLC2A4, TRIP10, KRAS, PCK1

miR-20b-5p

CRK, GRB2, PDE3A, PHKA2, PHKG1, PIK3R2, PRKAA2, PRKAR2A, MAPK9, PRKX, PPP1R3B, HK1, PCK1, SREBF1

miR-21-5p

PPP1R3A, PPP1R3D

miR-23a-3p

G6PC, IRS2, IKBKB, PIK3CB, FASN, PRKAG3

miR-24-3p

IKBKB, PIK3CB, PTPRF, SHC2, INPP5K, PHKG1, PRKAG3

miR-26a-5p

G6PC, MKNK2, GYS2, PPP1R3D, RHOQ, KRAS, PRKAG1, PYGL

miR-29a-3p

NRAS, EIF4E2, CALM3

miR-100-5p

MTOR

miR-103a-3p

FAS, RAPGEF1, PDE3B, ACACB, PHKAR1A, PRKCI, IRS2, LIPE, PRKC2, MAPK3,TRIP10, CBLC, CALML5

miR-125b-5p

PHKA1, RAF1, ACACB,FLOT2, HK2, EIF4E2, PHKG1

miR-126-3p

TSC1

miR-130b-3p

RPS6KB1, MAP2K1, SOS2, FLOT2, EXOC7, PHKG2, PIK3CA, PRKC2, TSC2, PRKAG3

miR-133a-3p

PRKAB1

miR-143-3p

FOXO1, KRAS, HK2, PHKG2, MAPK3, MAPK9, SREBF1

miR-145-5p

IRS1, IRS2, PIK3R5, PRKAG3

miR-181a-5p

NRAS, AKT3, SOCS4, HK2, PDE3B, PPP1R3C, PRKAR2A, MAPK1, PPP1R3D, PRKAA1

miR-195-5p

IKBKB, PHKA1, PRKAR1A, MAP2K1, RAF1, IRS4, MKNK1, EXOC7, FASN

miR-199a-5p

PRKX, PCK1, IRS1, SLC2A4, MAPK9, RHEB, PRKAR1A

miR-221-3p

AKT3, PIK3CD, MAPK10

miR-342-3p

PDPK1, INSR, PHKG2, EIF4E2, PIK3CD, RPS6KB2

miR-499a-5p

AKT2, CRK, KRAS, PIK3CD, PRKAR1A, SOS2

miR-574-3p

PRKCZ, HK1

Table 2b. A list of predicted targets of appropriate microRNAs dysregulated in whole peripheral blood of patients with a history of GDM in relation to type 1 diabetes mellitus pathway using miRWalk2.0 database

microRNA

Predicted targets

miR-1

CD28, LTA

miR-16-5p

HLA-DQA1

miR-17-5p

FASLG, CD28, HLA-DQA, GAD2, HLA-DPA1

miR-20a-5p

IL12A

miR-20b-5p

HLA-DOA, FASLG, CD28, HLA-DPA1, GAD2

miR-21-5p

HLA-DPB1, FASLG, IL12A

miR-23a-3p

IFNG

miR-24-3p

CD28, CD86, IFNG, FASLG, IL1B, HLA-DOA

miR-26a-5p

HLA-DPB1, HLA-DPA1, HLA-A, IFNG

miR-29a-3p

HLA-DQA2

miR-103a-3p

HLA-DPB1, CD80

miR-125b-5p

PRF-1

miR-130b-3p

HLA-DOA, HLA-DQB1, HLA-A, HLA-B, HLA-C, HLA-G

miR-133a-3p

HLA-DOA, CD28, GAD2, LTA

miR-143-3p

HLA-DOA, HLA-DPB1, HLA-DPA1, IFNG, CD28

miR-146a-5p

CD80, CD86, PRF1, ICA1, HLA-C, GAD2

miR-181a-5p

IL2, HLA-E, IL1A

miR-195-5p

HLA-DQA1

miR-199a-5p

G2MB, ICA1,TNF

miR-221-3p

HLA-DQA1, PTPRN

miR-342-3p

PTPRN2, HLA-A, HLA-F

Table 2c. A list of predicted targets of appropriate microRNAs dysregulated in whole peripheral blood of patients with a history of GDM in relation to type 2 diabetes mellitus pathway using miRWalk2.0 database

microRNA

Predicted targets

miR-16-5p

CACNA1E, IKBKB, IRS4

miR-17-5p

PIK3R2, MAPK9, HK1

miR-20a-5p

SLC2A4

miR-20b-5p

MAPK9, HK1

miR-23a-3p

IRS2, IKBKB, PIK3CB

miR-24-3p

IKBKB, PIK3CB, KCNJ11

miR-26a-5p

PRKCD

miR-29a-3p

CACNA1A, CACNA1B

miR-100-5p

MTOR

miR-103a-3p

CACNA1E, IRS2, PRKCZ, MAPK3

miR-125b-5p

HK2

miR-130b-3p

PIK3CA, PRKCZ

miR-143-3p

CACNA1A, HK2, PRKCE, MAPK3, MAPK9

miR-145-5p

IRS2, IRS1, PIK3R5

miR-146a-5p

PRKCE

miR-181a-5p

SOCS4, HK2, MAPK1

miR-195-5p

CACNA1E, IKBKB, IRS4

miR-199a-5p

IRS1, SLC2A4, MAPK9, PKM, TNF, CACNA1G

miR-221-3p

PIK3CD, MAPK10

miR-342-3p

CACNA1C, INSR, PIK3CD

miR-499a-5p

PRKCE, PIK3CD

miR-574-3p

PRKCZ, HK1

miRWalk database and the Predicted Target module were used to provide information on predicted interactions between appropriate microRNAs and specific genes involved in human biological pathways such as insulin signaling, type 1 diabetes mellitus, and type 2 diabetes mellitus.

Methods

4.6. Information on microRNA-gene-biological pathway interactions

MiRWalk database (available: http://www.umm.uni-heidelberg.de/apps/zmf/mirwalk/) and the Predicted Target module were used to provide information on predicted targets of those microRNAs that have been found to be dysregulated in whole peripheral blood of patients with a history of GDM. Only those predicted targets involved in particular human biological pathways (insulin signaling pathway, type 1 diabetes mellitus pathway, and type 2 diabetes mellitus pathway) are reported [183].

 miRWalk is a comprehensive database that provides information on microRNAs from human, mouse, and rat on their predicted and/or validated target genes. miRWalk2.0 not only documents miRNA binding sites within the complete sequence of a gene, but also combines this information with a comparison of binding sites resulting from 12 existing miRNA-target prediction programs (DIANA-microTv4.0, DIANA-microT-CDS, miRanda, mirBridge, miRDB4.0, miRmap, miRNAMap, DoRiNA, PicTar2, PITA, RNA22v2, RNAhybrid2.1, and Targetscan6.2) to build novel comparative platforms of binding sites for the promoter (4 prediction datasets), cds (5 prediction datasets), 5’- (5 prediction datasets) and 3’-UTR (13 prediction datasets) regions. Information on miRNA-target interactions on 2 035 disease ontologies (DO), 6 727 Human Phenotype ontologies (HPO) and 4 980 OMIM disorders is available. It provides possible interactions between microRNAs and genes associated with 597 KEGG, 456 Panther and 522 Wiki pathways.

Reviewer 3 Report

Diabetes mellitus and cardiovascular risk assessment in mothers with a history of gestational diabetes mellitus based on postpartal expression profile of microRNAs associated with diabetes mellitus, cardiovascular and cerebrovascular diseases

Ilona Hromadnikov et al. assessed a postpartal expression profile of microRNAs associated with cardiovascular/cerebrovascular disease 3-11 years after the delivery in whole peripheral blood of young and middle-aged mothers with a prior exposure to GDM. They aimed to identify highly risk group of mothers at risk of later development of diabetes mellitus and cardiovascular and cerebrovascular diseases, that would benefit from implementation of early primary prevention strategies and long-term follow-up.

The paper is scientifically interesting, but to make it more appealing, the authors should make a minor revision.

In particular, they should improve the Results:

- The Figure 1 is very long (panels A-Z). It isn’t of impact.

- The Figure 1 legend should be integrated with the description of the parts included in the figure (miRNA expression, ROC curve, Sensitivity, etc.).

- In the Table 1 there are several mistakes in miRNA names (i.e.: has-miR-103a-3     p??????, has-miR-125b-5       p??????, etc…). They should adjust the table and the spaces.

- The Table 2 is very long, the authors should divide it in more parts.

- Line 180: modify regadless with regardless

- Lines 306-307: which statistical tests used the authors? Parametric or not parametric? P-value or p-value? Please modify.

- Line 308: …a chi-square test.; eliminate the . please.

- Line 333: It's not nice to start a sentence with a number (3 mL), the authors should rephrase.

- Line 334: ingredients?????

- Line 377: The authors should verify, by the “Instructions for the authors” of the journal, if it is correct insert the Conclusions section after Materials and method.

Author Response

Response to Reviewer 3 Comments

Ilona Hromadnikov et al. assessed a postpartal expression profile of microRNAs associated with cardiovascular/cerebrovascular disease 3-11 years after the delivery in whole peripheral blood of young and middle-aged mothers with a prior exposure to GDM. They aimed to identify highly risk group of mothers at risk of later development of diabetes mellitus and cardiovascular and cerebrovascular diseases, that would benefit from implementation of early primary prevention strategies and long-term follow-up.

The paper is scientifically interesting, but to make it more appealing, the authors should make a minor revision.

In particular, they should improve the Results:

Point 1:  The Figure 1 is very long (panels A-Z). It isn’t of impact.

Response 1: We are aware of this situation. Newly, following suggestion of reviewer 2 the heat map summarizing the microRNA gene expression data was added.

Figure 1. Postpartal microRNA expression profile in mothers with a history of GDM pregnancies. MicroRNA gene expression data (2-DDCt) are visualised using the heat map. In this setting, each row represents a sample (GDM1 - GDM111, NP1 - NP89) and each column represents a microRNA gene. The colour and intensity of the boxes are used to represent changes of gene expression (2-DDCt). Green colour indicates up-regulation, and red colour indicates down-regulation.

Point 2: The Figure 1 legend should be integrated with the description of the parts included in the figure (miRNA expression, ROC curve, Sensitivity, etc.).

Response 2: Figure 1 (newly Figure 2) legend was expanded for more details as suggested.

(A-Z) Up-regulation of miR-1-3p, miR-16-5p, miR-17-5p, miR-20a-5p, miR-20b-5p, miR-21-5p, miR-23a-3p, miR-24-3p, miR-26a-5p, miR-29a-3p, miR-100-5p, miR-103a-3p, miR-125b-5p, miR-126-3p, miR-130b-3p, miR-133a-3p, miR-143-3p, miR-145-5p, miR-146a-5p, miR-181a-5p, miR-195-5p, miR-199a-5p, miR-221-3p, miR-342-3p, miR-499a-5p, and miR-574-3p was observed in mothers after GDM pregnancies when the comparison to the controls was performed using Mann-Whitney test. Receivers operating characteristic (ROC) curves were constructed to calculate the area under the curve (AUC), the best cut-off point (criterion), the sensitivity, specificity, likelihood ratio positive (LR+), and likelihood ratio negative (LR-) for particular microRNA. In addition, respective sensitivity at 90.0% specificity was reported for miR-1-3p (43.24%), miR-16-5p (20.72%), miR-17-5p (21.62%), miR-20a-5p (10.81%), miR-20b-5p (38.74%), miR-21-5p (28.83%), miR-23a-3p (17.12%), miR-24-3p (0.90%), miR-26a-5p (20.72%), miR-29a-3p (21.62%), miR-100-5p (13.51%), miR-103a-3p (30.63%), miR-125b-5p (9.91%), miR-126-3p (9.91%), miR-130b-3p (3.60%), miR-133a-3p (18.92%), miR-143-3p (13.51%), miR-145-5p (7.21%), miR-146a-5p (17.12%), miR-181a-5p (31.53%), miR-195-5p (16.22%), miR-199a-5p (15.32%), miR-221-3p (31.53%), miR-342-3p (1.8%), miR-499a-5p (28.83%), and miR-574-3p (10.81%).

Point 3: In the Table 1 there are several mistakes in miRNA names (i.e.: has-miR-103a-3     p??????, has-miR-125b-5       p??????, etc…). They should adjust the table and the spaces.

Response 3: We apologise for this inconvience. It was corrected as suggested.

Point 4: The Table 2 is very long, the authors should divide it in more parts.

Response 4: Table 2 was newly separated into 3 smaller subtables. Newly, Subtable 2a contains pre-existing cardiovascular risk factors before gestation. Subtable 2b contains data gained at follow-up. Subtable 2c contains the data gained during gestation.

Point 5: Line 180: modify regadless with regardless

Response 5: It was corrected as suggested.

Point 6: Lines 306-307: which statistical tests used the authors? Parametric or not parametric? P-value or p-value? Please modify. Line 308: …a chi-square test.; eliminate the . please.

Response 6: It was corrected as suggested.

Data are presented as mean±SE for continuous variables and as number (percent) for categorical variables. Continuous variables were compared using Mann-Whitney test (non-parametric test). p-value1: the comparison among normal pregnancies and GDM on diet ; p-value2: the comparison among normal pregnancies and GDM on the combination of diet and therapy, respectively. Categorical variables were compared using the chi-square test (non-parametric test). GA, gestational age; SBP, systolic blood pressure; DBP, diastolic blood pressure; CS, Caesarean section.

Point 7: Line 333: It's not nice to start a sentence with a number (3 mL), the authors should rephrase. Line 334: ingredients?????

Response 7: It was corrected as suggested.

cDNA (3 µL) was mixed with specific TaqMan MGB probes and primers (TaqMan MicroRNA Assay, Applied Biosystems, Branchburg, USA), and the components of the TaqMan Universal PCR Master Mix (Applied Biosystems, Branchburg, USA, no: 4318157).

Point 8: Line 377: The authors should verify, by the “Instructions for the authors” of the journal, if it is correct insert the Conclusions section after Materials and method.

Response 8: Conclusion section was placed correctly after Material and Methods section in original submission. We followed “Instructions for the authors” during submission process of original manuscript.

Round 2

Reviewer 2 Report

Authors made changes that improved the quality of the manuscript.

However, the main findings of the whole paper are weak.

It is a very descriptive study in which only a list of deregulated microRNA is provided. Reviewer feels that a list of deregulated microRNAs is not enough for a peer review paper in a journal of such quality.